# A new investigation of nonalcoholic fatty liver disease: Effects of hypoxia on mitochondrial function and lipid droplet autophagy

Meiyuan Tian[1,2], Yaogang Zhang[1,2], Zhe Liu[1,2], Na Zhao[1,2], Dengliang Huang[1,2], Jing Hou[1,2], Li Sun[1,2], Yuan Jiang[1,2], Guangcun Zhang[1,2], Yanyan Ma[1,2]*

1 Central Laboratory, Affiliated Hospital of Qinghai University in Qinghai province, Xining, Qinghai Province, China, 2 Qinghai Province Research Key Laboratory of Echinococcosis, Qinghai University Affiliated Hospital, Xining, Qinghai Province, China

* mayanyan_research@qhu.edu.cn

## Abstract

An expanding body of research has highlighted the intimate connection between nonalcoholic fatty liver disease (NAFLD) and the dysregulation of hepatic lipid droplet autophagy as well as mitochondrial function. Nevertheless, the intricate interplay between lipid droplet autophagy and mitochondrial function under hypoxic conditions remains largely uncharted territory. Constructing NAFLD mouse models at altitudes of 2200 meters and 4500 meters and simultaneously culturing hepatocytes under oxygen concentrations of 21% and 1%, with the addition of oleic acid to induce lipid accumulation. A comprehensive evaluation of the NAFLD mice at different altitudes was conducted, including a combination of NMR, PAS, and Oil Red O staining, immunofluorescence, qPCR, flow cytometry, ELISA, electron microscopy, and cellular energy metabolism experiments. The high-altitude, high-fat diet group exhibited a reduction in lipid deposition and glycogen content, an increase in lipid droplet autophagy, and a decrease in mitochondrial damage and inflammatory injury when compared to the moderate-altitude, high-fat diet group. The 1% $O_2$ + oleic acid group exhibited enhanced lipid droplet autophagy and increased cellular adaptation in comparison to the 21% $O_2$ + oleic acid group. This study revealed that hypoxic conditions enhanced lipid droplet autophagy, reduced glycolipid accumulation, and alleviated mitochondrial damage in NAFLD mice.

## Introduction

Nonalcoholic fatty liver disease (NAFLD) affects a quarter of the general population and is prevalent among morbidly obese patients worldwide [1].The disease is distinguished by the accumulation of lipids in hepatocytes and is a dynamic process. As lipid deposition increases, it exacerbates hematologic damage to hepatocytes, inducing oxidative stress, mitochondrial damage, and the release of inflammatory factors,

**Data availability statement:** All relevant data are within the manuscript and its Supporting Information files.

**Funding:** This study was financially supported by the Qinghai University Research Capability Enhancement Project (2025-KTSQ-05) and the Qinghai Province "Kunlun Talent – High-End Innovation and Entrepreneurship Program" Direct-Certification Track for Top-Tier Talents. The funders had no role in study design, data collection and analysis, the decision to publish, or manuscript preparation.

**Competing interests:** No.

which ultimately contribute to the aggravation of hepatocyte damage [2,3]. Recent studies have shown that risk factors for NAFLD include dysregulation of autophagy in lipid droplets, in addition to metabolic abnormalities, genetics, inflammatory damage, intrinsic immunity and mitochondrial dysfunction [4]. The intracellular lipid degradation process known as lipid droplet autophagy plays a pivotal role in maintaining lipid homeostasis and energy balance [5]. In a hypoxic environment, the activation of AMP-activated protein kinase (AMPK) leads to the suppression of the mammalian target of rapamycin (mTOR) pathway. Consequently, this suppression promotes autophagy, which in turn augments the removal of lipid droplets [6]. The ATGL and patatin-like phospholipase domain-containing protein 8 (PNPLA8) serve as selective autophagy receptors for lipid engulfment, thereby facilitating lipolysis and β-oxidation of free fatty acids (FFAs) [7]. During the process of lipid droplet autophagy, lipid droplet-associated proteins perilipin 2 (PLIN2) and perilipin 3 (PLIN3) are degraded through chaperone-mediated autophagy (CMA) by a synergistic effect of Hsc70 and lysosome-associated membrane protein 2A (LAMP-2A) receptors [8]. Nevertheless, lipid droplet autophagy and mitochondrial function have been the subject of considerable investigation, yet their precise role in the pathogenesis of NAFLD under hypoxia remains uncertain. Furthermore, evidence suggests that hypoxia may play a role in alleviating liver triglyceride accumulation, necrotizing inflammation, and fibrosis during NAFLD [9]. Studying the lipid droplet autophagy and mitochondrial function with NAFLD in a hypoxic environment is significant for preventing and treating NAFLD.

Lipid droplet autophagy is one of the important mechanisms for maintaining intracellular lipid homeostasis, and its dysregulation may lead to abnormal lipid accumulation in hepatocytes [10]. Impaired lipid droplet autophagy can cause cells to become sensitive to death stimuli and may contribute to the development of a variety of diseases, including NAFLD and metabolic syndrome [11]. Significant advancements have been made in this field. However, the specific role of high-altitude hypoxic environments on mitochondrial function and lipid droplet autophagy in NAFLD is unknown. The role of hypoxia in the development of NAFLD is an emerging area of research. It is important to investigate whether obesity-related NAFLD is affected by altitude [12]. Understanding why the risk of NAFLD is lower in high-altitude areas with hypoxia conditions than in plain areas is significant for comprehending abnormal lipid metabolism in NAFLD under hypoxia conditions [13]. Hypoxia and lipid metabolism are known to be linked [14], although the exact mechanism remains unclear.

Consequently, we constructed a mouse model of NAFLD under hypoxia for validation purposes and explored the level of autophagy in lipid droplets of NAFLD mice and lipid-accumulating hepatocytes under hypoxia, as well as the mitochondrial function of lipid-accumulating hepatocytes under hypoxia. The findings provide new clues for preventing NAFLD under hypoxia.

## Materials and methods

### Materials

Antibodies APOA4 (Proteintech, #17996-1-AP), PPARα (Proteintech, #66826-1-Ig), DGAT2 (Proteintech, #17100-1-AP), ATGL (Proteintech, #55190-1-AP) were

purchased from Proteintech (Wuhan, China). PAS (Servicebio, GP1039) and oil red O (Servicebio, GP1067) were purchased from Servicebio (Wuhan, China). SYBR-Green (0491850001) was purchased from Roche (Basel, Switzerland). The TSA 4 color staining kit (Cat# TGT4C50) was purchased from Tissue Gnostics (Beijing, China). The cell mitochondria pressure test kit (Cat# 103015-100) was purchased from Agilent (Beijing, China). The mitochondrial membrane potential (JC-1, M8650) and mitochondrial membrane permeability transition pore (mPTP, BB-48122) test kits were purchased from Beijing Solarbio Science & Technology Co., Ltd. (Beijing, China).The reactive oxygen species (ROS) detection kit (S0033S) was purchased from Beyotime. The IFN-γ test kit was obtained from Songon Biotech (Shanghai, China). The alanine aminotransferase (ALT, A110-1-1) test kit was purchased from Nanjing Jiancheng Bioengineering Institute (Nanjing, China).

## Animal experiments

C57/BL6J male mice (n = 60; weight = 20 ± 1.87 g; 8 weeks old) were purchased from Beijing Vital River Laboratory Animal Technology Co., Ltd. (Beijing, China). C57/BL6J mice were adaptively fed for 1 week and then randomly divided into four groups, 15 in each group. Control groups were raised at Xining (altitude, 2200 m; Xining, Qinghai, China), fed a normal diet (ND-2200 m) and a high-fat diet (HFD-2200 m). Hypoxia groups were raised in a simulated low-pressure oxygen chamber at an altitude of 4500 m (simulated altitude, 4500 m; Xining, Qinghai, China), fed a normal diet (ND-4500 m) and a high-fat diet (HFD-4500 m). Continuous administration for 19 weeks. The animal low-pressure oxygen cabin parameters were set to simulate an altitude of 4,500 m with a lifting speed of 10 m/s and a cabin pressure of 45 KPa. ND formula: crude protein (19.24%), crude fat (4.6%), crude ash (6.2%), water (9.3%), calcium (0.959%), phosphorus (0.67%), vitamin A ($6.69 \times 10^3$ IU/kg), manganese (90 mg/kg), zinc (50 mg/kg), potassium ($9.1 \times 10^3$ mg/kg), 17 total amino acids (17.20%); HFD formula: 77.3% normal diet+ 10% egg yolk powder+ 10% lard+ 2.5% cholesterol+ 0.2% sodium cholate [15]. Two altitudes were selected for the study: 2200 meters, which represents a moderate altitude, and 4500 meters, which represents a high altitude. These altitudes were selected for the experiments primarily based on their capacity to replicate the physiological challenges that humans may face in diverse high-altitude environments, including low oxygen, high cold, and elevated ultraviolet radiation [16]. The impact of these environmental factors on the human body, particularly with regard to lipid metabolism and liver function, represents a crucial area of investigation in the context of metabolic diseases such as NAFLD. Following the conclusion of all experimental procedures, the mice were euthanised through an intraperitoneal injection of 60 mg/kg sodium pentobarbital, followed by cervical dislocation. The sodium pentobarbital was obtained from Wuhan Jinnuo Chemical Co., Ltd. (China). The presence of no respiration and no cardiac activity confirmed the death of the subject. All procedures involving animals were conducted in strict accordance with the guidelines approved by the Medical Ethics Committee of Qinghai University Affiliated Hospital (Xining, China), under the ethical code P-SL-202298.

## 7T magnetic resonance imaging

A small animal Magnetic Resonance Imaging (MRI, Pharma Scan 70/16 us, Germany) examination was performed on four groups of mice to observe the liver imaging changes. The specific methods were as follows: the mice were fasted for 12 hours prior to the examination, then placed into the anesthesia induction box. The concentration of isoflurane was adjusted from 1.0% to 2.0%. The mice were initially placed in a general anesthetic state via inhalation of 0% isoflurane, which was then maintained continuously. Subsequently, the morphology, size, structure, and subcutaneous fat of the livers were examined using hepatic T2 Spectral Presaturaton with Inversion Recovery, following stabilization of the heart rate and respiration. The images were then captured.

## Sample collection

Following a 19-week period of management, the mice were sacrificed under deep anesthesia. The liver tissue was harvested and divided into two portions. One portion was stored in 4% paraformaldehyde, while the other was stored at −80 °C after flash freezing in liquid nitrogen.

## Periodic acid-Schiff and oil-red O staining

To analyze the structural changes of hepatocytes and liver lipid disposition, liver tissue specimens were fixed with 4% paraformaldehyde and stained with periodic acid-Schiff (PAS, Servicebio, GP1039) and oil red O (Servicebio, GP1067). The images were acquired using the TISSUE GNOSTICS Analysis System (TISSUE FAXS-S Plu, ZEISS Axio Imager Z2, Austria). The analysis of lipid droplets, mean distance between nuclei and nuclei, as well as glycogen contents were conducted using the Strata Quest software (TISSUE GNOSTICS, Austria).

## Cell culture

The alpha mouse liver 12 (AML-12) cell line was purchased from Procell (BNCC338281,Wuhan, China). AML-12 cells were cultured in Dulbecco's modified Eagle's medium (DMEM) containing 10% serum. In the cell culture experiments, two oxygen concentrations were selected for the cultures: 1% and 21%. These concentrations were chosen for their ability to represent extreme hypoxic conditions and normal physiological oxygen concentrations, respectively. By comparing the changes in the growth status, metabolic levels, and gene expression profiles of cells under these two conditions, a deeper understanding of the effects of the hypoxic environment on cellular functions can be gained. Hepatocytes were cultured in 21% and 1% $O_2$ incubators, respectively. Hepatocytes were treated with 60 μg/mL oleic acid for 72 h to simulate lipid accumulation [15].

## Flow cytometry experiment

Flow cytometry was used to detect the mitochondrial membrane potential (JC-1, M8650), mitochondrial membrane permeability transition pore (mPTP, BB-48122) and reactive oxygen species (ROS, S0033S) according to the instructions of the kit. Hepatocyte single-cell suspensions were obtained by grinding mouse livers, and 400,000 hepatocytes were counted. The hepatocytes were then incubated with the fluorescent probes in the JC-1 and mPTP kits to detect the mitochondrial membrane potential and the degree of openness of the mitochondrial membrane permeability transition pore, in accordance with the methods described therein. Positive (untreated) and negative controls (FCCP/CCCP) should be set up. Cells are gated through FSC/SSC to select the cell population and exclude dead cells. Fluorescence intensity is analyzed to determine the proportion of healthy and damaged mitochondria. For detecting the mitochondrial permeability transition pore (MPTP), Calcein-AM and cobalt ions are used in combination. Fluorescence intensity changes are analyzed in the FITC channel, and positive and negative controls were set to verify MPTP opening. In experiments, dye concentrations need to be optimized to ensure data reproducibility. Software such as FlowJo can be used to analyze distribution and significant differences.

## The detection of biochemical indicators and inflammatory factors

The liver tissue levels of ALT were assayed using the ALT kits, according to the manufacturer's instructions. The serum levels of IFN-γ were assayed using ELISA kits (D721025-0096), following the methods of Sangon Biotech.

## Cell mitochondria pressure measurement

Collect oleic acid processing 72 h of hepatocytes, $1 \times 10^4$ hepatocytes inoculated respectively in the microporous plate, according to the Seahorse XF cell mitochondria pressure measurement kit instructions, through the cell energy metabolism to test the extracellular acidification rate (Extracelluar acidification rate, ECAR) and cellular oxygen consumption rate (OCR) levels.

## Immunofluorescence

The frozen sections were infiltrated with 1% NP-40 for 5 min and blocked with 5% BSA serum albumin at room temperature for 2 h to eliminate nonspecific staining. The primary antibodies, APOA4 (1:100), PPARα (1:200), DGAT2 (1:200),

and ATGL (1:200), were incubated at 37 °C for one hour. After 4 rounds of staining using the TSA 4 color staining kit, the nuclei were stained and the slides were sealed. Images were acquired and analyzed using a TISSUE GNOSTICS Analysis System (TISSUE FAXS-S Plu, ZEISS Axio Imager Z2, Austria).

### RT-PCR

The total RNA was extracted using Trizol reagent (Thermo Fisher, 15596026), following the manufacturer's instructions. Subsequently, the total RNA was reverse transcribed into complementary DNA (TransGen Biotech, AT311−02) and amplified by qPCR Master Mix (Roche, 0491850001) using the Real-time PCR system (Roche, Switzerland). The $2^{-\Delta\Delta CT}$ method was used to determine all samples. The primer sequences are listed in Table 1.

### Statistical analysis

The results were analyzed using the SPSS software (version 28.0, USA). Experimental data were obtained from repeated measurements and analyses of various experiments. Quantitative data are presented as mean ± standard deviation (SD). Two-way analysis of variance (ANOVA) and Tukey's multiple comparison tests were used for statistical analysis. An unpaired t-test was used to compare the two groups. Statistical significance was set as $P < 0.05$.

## Results

### Effects of hypoxia on lipid deposition and glycogen storage in NAFLD mice

Given the high degree of homology between mouse and human genes, we employed a NAFLD mouse model that was developed through the administration of a high-fat diet for 19 weeks. To objectively evaluate the success of establishing the NAFLD mouse model, small animal MRI monitored the T2 Spectral Presaturation with inversion recovery of the livers of the four groups of mice. Additionally, we analyzed the lipid deposition and the content of glycogen and polysaccharide substances by PAS and oil red O staining to observe the degree of hepatic steatosis in the four groups of mice.

The results of the magnetic resonance imaging (MRI) scans indicated that the subcutaneous fat accumulation increased with the duration of the modeling process. Additionally, the subcutaneous fat in the livers of mice in the HFD-4500 m group was observed to be less dense than that in the HFD-2200 m group (Fig 1A and 1B). The results of the PAS and oil red O staining procedures indicate that in the high-fat diet group, the hepatocytes of the mice were observed to be swollen, and fat vacuoles were present within the cytoplasm (Fig 1C and 1G). The liver tissue exhibited the presence of both large and small oil droplets, which were observed to be dispersed throughout the organ (Fig 1D). The percentage of lipid droplets in oil red O-stained sections and the percentage of polysaccharide material in PAS sections were analyzed

**Table 1. The sequence of the primers.**

| Gene | primer sequence (5'-3') |
|---|---|
| Apoa4 | forward primer: CAGAAGACGGATGTCACTCAGC |
| | reverse primer: AGCTGTACGACAAAGGGCACCA |
| Pparα | forward primer: TGCCTTCCCTGTGAACTGAC |
| | reverse primer: TGGGGAGAGAGGACAGATGG |
| Dgat2 | forward primer: ATTTGACTGGAACACGCCCA |
| | reverse primer: ACATCAGGTACTCGCGAAGC |
| 18srRNA | forward primer: GTAACCCGTTGAACCCCATT |
| | reverse primer: CCATCCAATCGGTAGTAGCG |
| Atgl | forward primer: GGAACCAAAGGACCTGATGACC |
| | reverse primer: ACATCAGGCAGCCACTCCAACA |

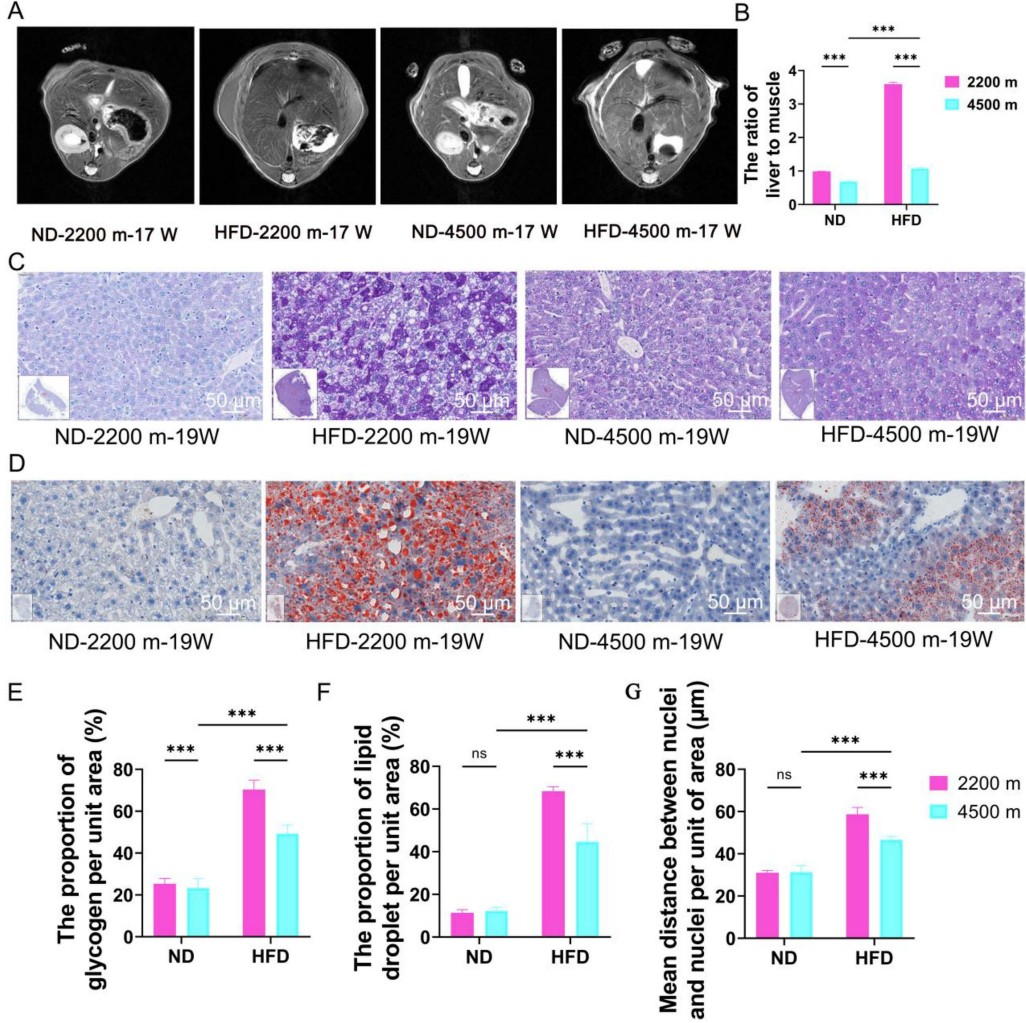

**Fig 1. Effects of hypoxia on lipid deposition and glycogen storage in NAFLD mice.** (A) MRI was employed to monitor the hepatic image of the NAFLD mouse model.(B) The liver-to-muscle signal intensity ratio (LMSIR) measured by NMR is used for objective assessment of liver fat content, quantifying the NMR signal characteristics of liver tissue relative to muscle, and serves as an important parameter reflecting liver steatosis (n = 4 per group).(C) PAS staining was conducted on liver tissue from the NAFLD mouse model at the 19th week (Glycogen appears purplish-red, and the cell nuclei are light blue.). The scale bar represents 50 μm.(D) Oil Red O staining was conducted on liver tissue from the NAFLD mouse model at the 19th week (Lipid droplets appear orange-red to bright red, and the cell nuclei are light blue.). The scale bar represents 50 μm.(E) Statistical results of glycogen percentage in liver tissue in mice (n = 3 per group).(F) Statistical results of lipid droplet percentage in liver tissue in mice (n = 3 per group).(G) Statistical results of the mean distance between the nuclear and the nuclear per-unit area (n = 3 per group).*$P < 0.05$; **$P < 0.01$; ***$P < 0.001$, ns, no significance.

using Strata Quest software, which revealed a reduction in lipid deposition and glycogen percentage in the HFD-4500 m group compared to the HFD-2200 m group (Fig 1E and 1F).

## Hypoxia attenuates hepatocyte injury in the NAFLD mouse model

Abnormalities in lipid metabolism have been linked to a range of issues, including mitochondrial dysfunction, excessive oxidative stress, endoplasmic reticulum stress, inflammation, and fibroplasia. These factors contribute to an increased susceptibility of the liver to the development of metabolic diseases. To further examine the impact of a hypoxic

environment on mitochondria in mice from a high-fat diet group, hepatocyte suspensions were obtained by grinding liver tissue from four groups of mice at 19 weeks. Flow cytometry was employed to analyze mitochondrial membrane potential (JC-1 fluorescent probe staining), mitochondrial membrane permeabilization transition pore (mPTP) activity and ROS level. The high-fat diet resulted in a notable increase in mitochondrial membrane potential damage in mice, whereas the degree of membrane potential damage exhibited a decline with elevation, indicating that fat accumulation caused damage to hepatocyte mitochondria, which was observed to be ameliorated in the low-oxygen environment at high altitude (Fig 2A and 2C). The mPTP results demonstrated a reduction in the green fluorescence of calcein in mitochondria in response to the high-fat diet indicated that the opening of the mPTP in the mitochondria was increased, suggesting that lipid accumulation damaged mitochondria. In contrast, at high altitude and low oxygen, the opening of mitochondria was decreased (Fig 2B and 2D). ROS results showed that oxidative damage in hepatocytes increased with the increase in high-fat diet, but this damage was mitigated with increasing altitude (Fig. 2C and 2F). Additionally, the impact of hypoxia on the inflammatory factor IFN-γ was evaluated via ELISA. The findings revealed that the serum IFN-γ concentration was diminished in the HFD-4500 m group of mice relative to the HFD-2200 m group (Fig 2E). The levels of the alanine aminotransferase (ALT) in the liver tissue of the HFD-4500 m group were lower than those in the HFD-2200 m group (Fig 2F). These outcomes suggest that high-altitude hypoxia may attenuate mitochondrial damage,oxidative damageand inflammatory injury in hepatocytes.

### Effect of hypoxia on mitochondrial function in lipid-accumulating hepatocytes

The available evidence increasingly suggests that hepatocellular mitochondrial dysfunction plays a crucial role in the onset and pathogenesis of NAFLD [17]. Mitochondrial function is believed to be instrumental in regulating the hepatic hypoxic response [18]. Alterations in autophagy and mitochondrial function are thought to be pivotal in linking tissue hypoxia, oxidative stress, and lipid metabolism in the liver [19]. Nevertheless, the effects of hypoxia on autophagy levels, mitochondrial function, and lipid deposition in NAFLD have not been extensively investigated. The results of mitochondrial function tests showed reduced proton leakage and enhanced spare respiration, and non-mitochondrial respiration, ATP production in lipid-accumulating hepatocytes under hypoxia compared to lipid-accumulating hepatocytes under normoxia, suggesting reduced mitochondrial damage and increased cellular adaptation (Fig 3A–3G).

### Enhanced autophagy in lipid droplets of NAFLD mice and lipid-accumulating hepatocytes under hypoxia

To gain further insight into the extent of lipid droplet autophagy in mice in the high-fat group under hypoxic conditions, we employed immunofluorescence to label the lipophagy-related proteins ATGL, APOA4, PPARα, and the lipid droplet fusion-associated protein DGAT2. Subsequently, we conducted a detailed analysis of the positive rate. The results demonstrated that, in comparison to the HFD-2200 m group, the HFD-4500 m group exhibited a greater number of ATGL, APOA4, and PPARα-positive cells, a reduced number of DGAT2-positive cells, and a higher prevalence of double-positive cells for APOA4 and PPARα (Fig 4A–4F). In accordance with the aforementioned findings, the qPCR results demonstrated elevated mRNA levels of *Atgl, Apoa4*, and *Ppara* and diminished mRNA levels of *Dgat2* in the HFD-4500 m group relative to the HFD-2200 m group (Fig 4G–4J). The aforementioned results indicate that lipid accumulation under hypoxic conditions enhances autophagy in lipid droplets and that APOA4 is driven by PPARα. Lipid droplet autophagy represents a novel form of selective autophagy, a lipid metabolic process also known as "lipophagy". This process involves the autophagic degradation of intracellular lipid droplets, which plays a role in regulating intracellular lipid storage, intracellular free lipid levels (e.g., fatty acids), and energy homeostasis [20]. In addition, we analyzed at the cellular level. The structures of autophagy vesicles, lipid droplets, and autophagy lysosomes were observed in four groups of cells by electron microscopy. The number of autophagy vesicles and autophagy lysosomes was found to be greater in the 1% $O_2$+oleic acid group than in the 21% $O_2$+oleic acid group (Fig 4K). This finding suggests that autophagy of lipid droplets in lipid-accumulating hepatocytes is enhanced under hypoxia.

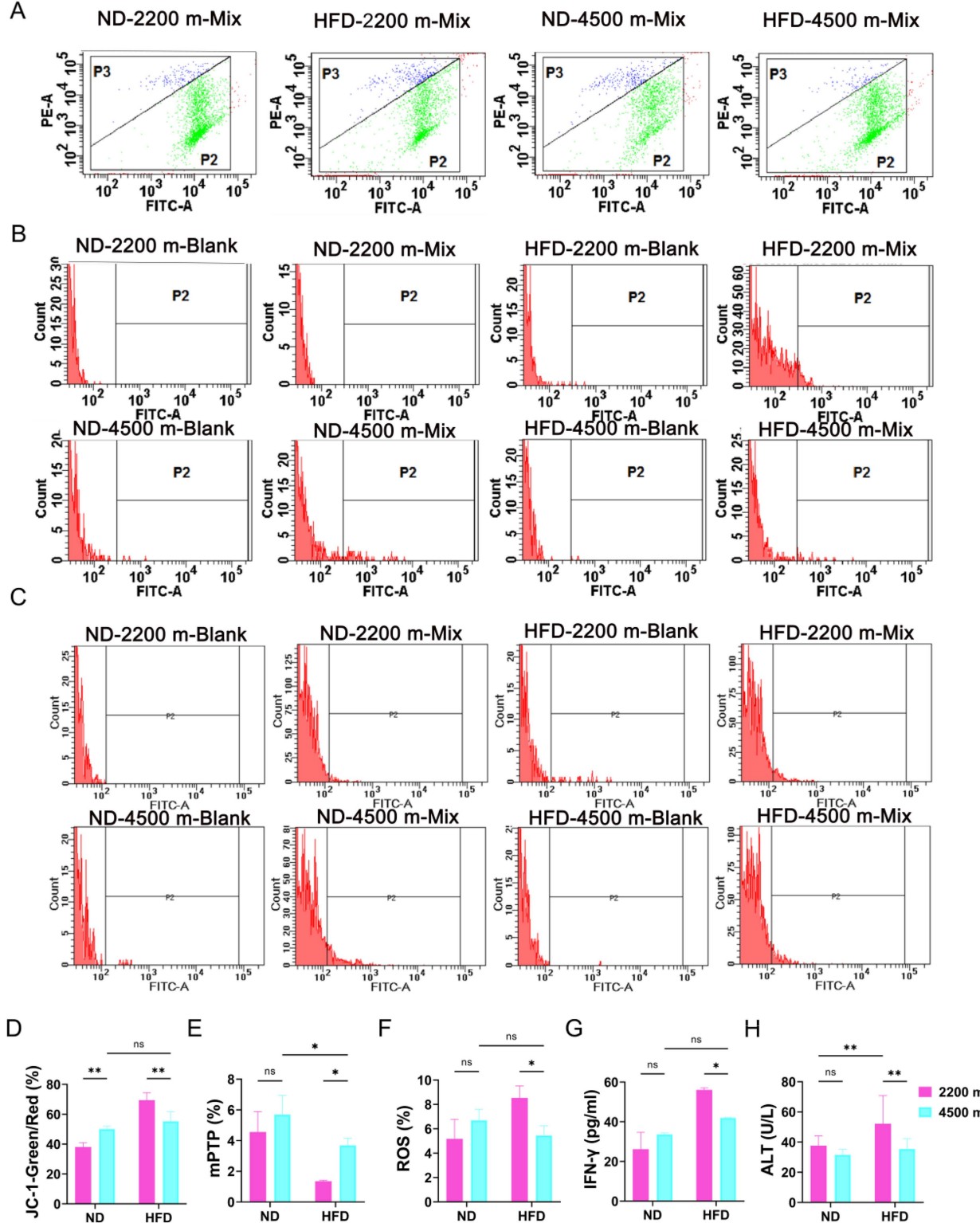

**Fig 2. Effect of hypoxia on hepatocyte injury in NAFLD model mice.** (A) Analysis of mitochondrial membrane potential in hepatocytes in vivo. (B) Analysis of hepatocyte mitochondrial membrane permeability transition pore in vivo (In normal cells, only Calcein in the cytoplasm was quenched; However, MPTP was opened to a certain extent in injured cells, at which time $CoCl_2$ could enter the mitochondria. Therefore, in addition to all Calcein in the cytoplasm, Calcein in the mitochondria was also quenched to varying degrees in injured cells). (C) Analysis of hepatocyte ROS level in vivo. (D)

Statistical results of cells with decreasing mitochondrial membrane potential (n = 3 per group). (E) Statistical results of the opening of the mitochondrial membrane permeability transition pore (n = 3 per group). (F) Statistical results of ROS level in vivo (n = 3 per group). (G) Analysis of NAFLD model mice liver tissue content of IFN-γ (n = 9 per group). (H) Analysis of alanine amino transferase content in NAFLD mice liver tissue (n = 3 per group). ns, no significance; *P < 0.05; **P < 0.01; ***P < 0.001.

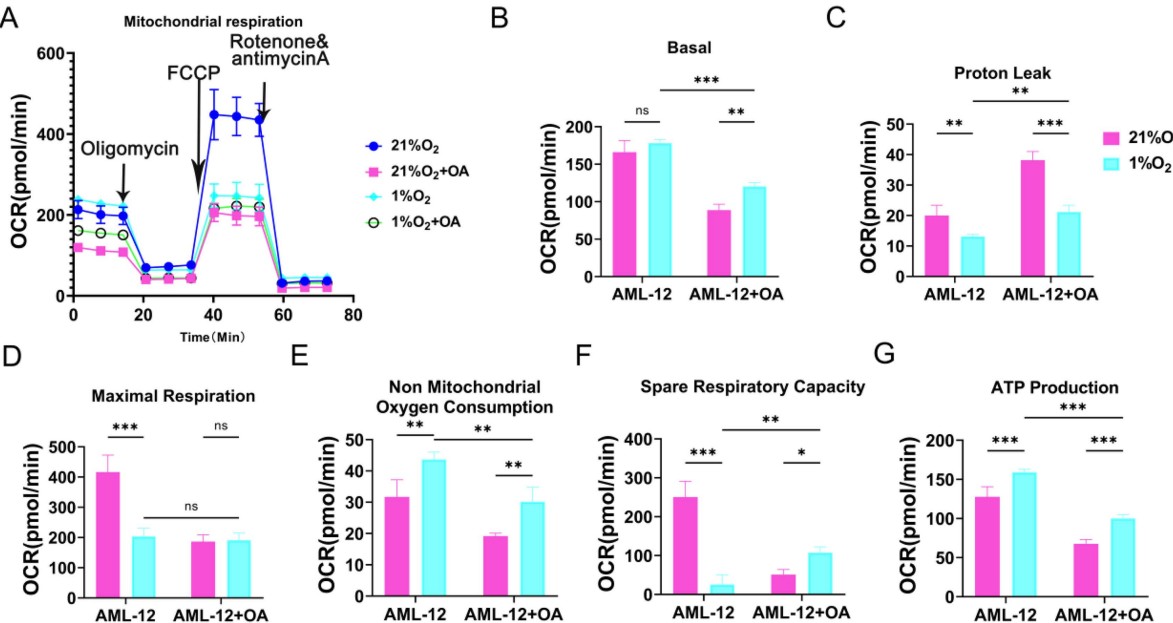

**Fig 3. Enhanced adaptive capacity of lipid-accumulating hepatocytes under hypoxia.** (A) Hepatocyte mitochondrial respiratory oxygen consumption: The oxygen utilization efficiency of hepatocyte mitochondria during ATP production.(B) Hepatocyte basal respiratory oxygen consumption: The total oxygen consumption of hepatocytes under basal conditions, including mitochondrial respiration and other non-mitochondrial oxygen-consuming processes (n = 3 per group).(C) Hepatocyte proton leak: The process by which protons leak back into the mitochondrial matrix through non-ATP synthase pathways across the inner mitochondrial membrane, resulting in the consumption of some oxygen that is not used for ATP synthesis (n = 3 per group). (D) Hepatocyte maximal respiratory oxygen consumption: The maximum oxygen consumption rate achievable by mitochondria when fully activated (n = 3 per group).(E) Hepatocyte non-mitochondrial respiratory oxygen consumption: The amount of oxygen consumed by cellular organelles or metabolic pathways other than mitochondria (n = 3 per group).(F) Hepatocyte respiratory potential: The transmembrane potential difference formed during electron transfer in the mitochondrial respiratory chain, which serves as the driving force for ATP synthesis (n = 3 per group).(G) Hepatocyte ATP production: The total amount of ATP produced by hepatocytes through mitochondrial oxidative phosphorylation or other metabolic pathways, reflecting the cell's energy output capacity (n = 3 per group).*P < 0.05; **P < 0.01; ***P < 0.001, ns, no significance.

## Discussion

Although NAFLD is a serious global public health problem, its mechanism under hypoxia is unclear [21]. A number of physiological and biochemical indices of plateau residents differ from those of plains residents, which presents a challenge for the diagnosis and therapeutic decision-making of diseases affecting plateau residents [22]. It is thus imperative to elucidate the regulatory mechanism of NAFLD in the context of high altitude and low oxygen levels, with a view to facilitating the scientific prevention and control of NAFLD in the plateau region. The present study revealed that, in comparison to the HFD-2200 m group, the HFD-4500 m group exhibited a notable reduction in subcutaneous fat, glycogen content, and lipid deposition (Fig 1). The weekly feed intake for the four groups of mice was standardized. The results indicate that the effect of hypoxia on the lipid deposition of mice fed a high-fat diet may be attributable to hypoxia rather than a reduction in food intake. Studies have shown that metabolic processes in animals show long-term changes in response to hypoxia. Chronic hypoxia improved the body weight and insulin resistance in NASH mice [23]. In our study,

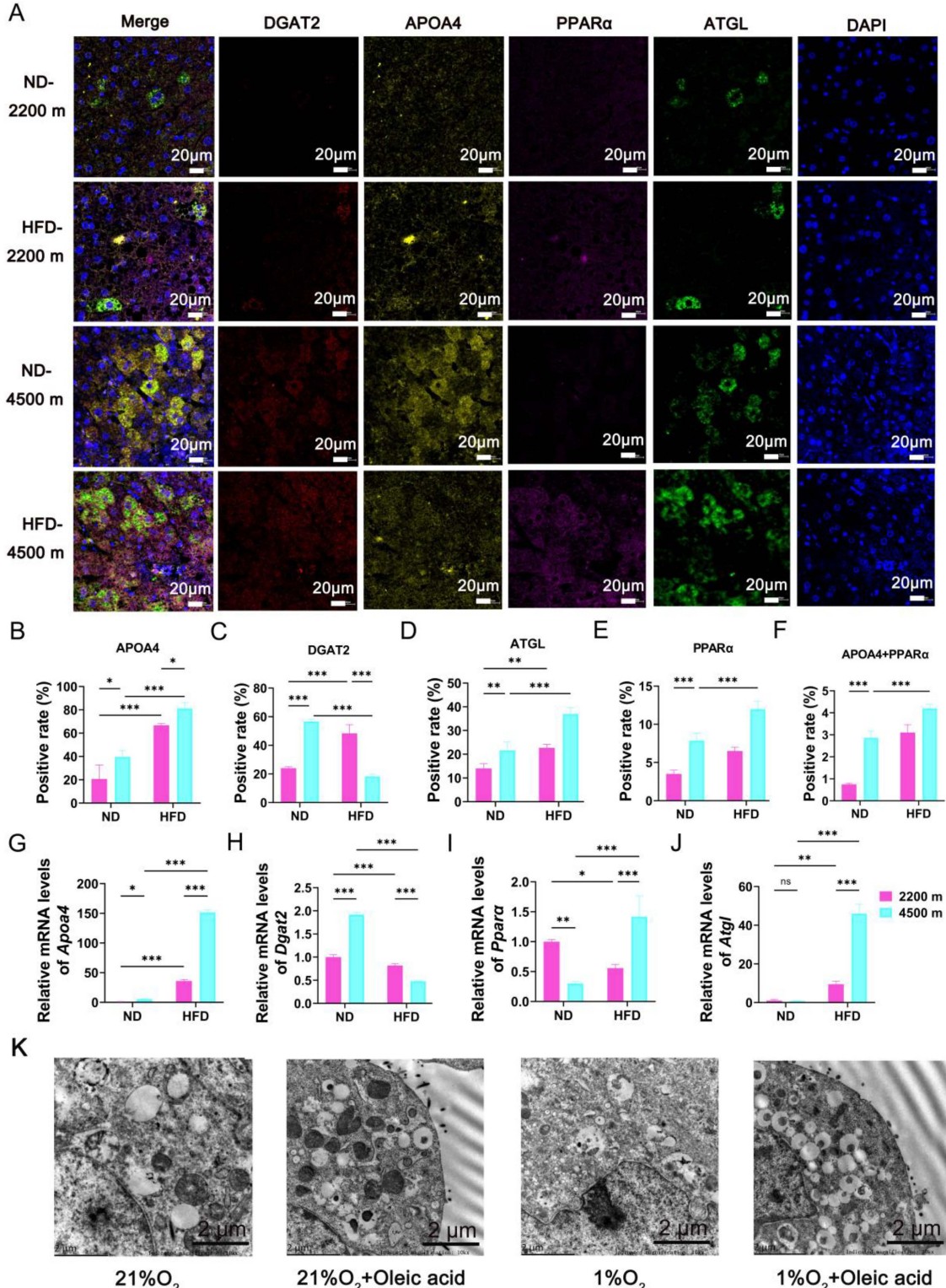

**Fig 4. Enhanced autophagy in lipid droplets of NAFLD mice and lipid-accumulating hepatocytes under hypoxia.** (A) TSA staining of liver tissue in NAFLD mice (n = 3 per group).(B) Analysis of the ratio of APOA4 protein-positive cells to all cells per unit area (n = 3 per group).(C) Analysis of the ratio of DGAT2 protein-positive cells to all cells per unit area (n = 3 per group).(D) Analysis of the ratio of ATGL protein-positive cells to all cells per unit

area (n = 3 per group).(E) Analysis of the ratio of PPARα protein-positive cells to all cells per unit area (n = 3 per group).(F) Analysis of the ratio of cells with double-positive APOA4 protein and PPARα protein to all cells per unit area (n = 3 per group).(G) Apoa4 mRNA relative expression in vivo (n = 3 per group).(H) Dgat2 mRNA relative expression in vivo (n = 3 per group).(I) Pparα mRNA relative expression in vivo (n = 3 per group).(J) Atgl mRNA relative expression in vivo (n = 3 per group).(K) Electron microscopy observed autophagy in lipid droplets in vitro (n = 3 per group); the scale bar represents 5 μm.*P < 0.05; **P < 0.01; ***P < 0.001;ns, no significance.

the association of abnormal lipid metabolism with oxidative stress may also be modulated by the hypoxic environment. Together, oxidative stress and aberrant lipid metabolism drive the development of multiple diseases. Mitochondrial homeostasis links redox reactions to lipid metabolism. Hypoxia, oxidative stress, and inflammatory conditions all lead to increased levels of reactive oxygen species (ROS), which exacerbate lipid peroxidation [24].

A notable characteristic of NAFLD is the accumulation of lipid droplets in hepatocytes [25]. Abnormal mitochondrial function and disturbances in lipid droplet autophagy are considered to be significant contributors to the development and progression of NAFLD [26]. Hepatic hypoxia is a prominent feature of numerous pathological conditions and is linked to the development of alcoholic and nonalcoholic fatty liver disease [27]. The present study demonstrated that mitochondrial damage was attenuated in NAFLD mice under hypoxic conditions (Fig 2). Mitochondrial dysfunction is a key factor in the pathogenesis of non-alcoholic fatty liver disease (NAFLD), affecting energy metabolism and lipolysis in hepatocytes [17,28]. Furthermore, our research revealed that lipid-accumulating hepatocytes demonstrate enhanced adaptation under hypoxic conditions (Fig 3). It has been demonstrated that the restoration of mitochondrial autophagy in hepatocytes can enhance mitochondrial function and structure, thereby mitigating the progression of NAFLD [29]. Studies have indicated that augmented mitochondrial autophagy can facilitate the degradation of damaged mitochondria, thereby restoring mitochondrial function and accelerating mitochondrial fatty acid oxidation [30]. This process is conducive to the reversal of hepatic fatty acid accumulation and the improvement of insulin resistance [31]. Our research revealed that the HFD-4500 m group demonstrated augmented lipid droplet autophagy in hypoxic conditions (Fig 4). Impaired lipophagy can render cells susceptible to death stimuli, which may contribute to the development of a variety of diseases, including NAFLD and metabolic syndrome [32,33]. Our study challenges the conventional view that hypoxia always exacerbates mitochondrial dysfunction in the context of NAFLD. Instead, our findings suggest that hypoxia may have a protective effect on mito-chondria in NAFLD mice, potentially through the induction of mitochondrial autophagy. This process can facilitate the degradation of damaged mitochondria, restore mitochondrial function, and accelerate mitochondrial fatty acid oxidation. Differences in the NAFLD mouse models used (e.g., diet-induced vs. genetic models) may lead to varying responses to hypoxia compared to previous studies. Variations in the timing, duration, and severity of hypoxia exposure across stud-ies can influence whether observed effects represent adaptation versus early injury or stress. Our focus on lipid droplet autophagy suggests hypoxia might activate protective pathways to mitigate metabolic stress, differing from traditional views of direct mitochondrial damage and lipid accumulation.As a result, it may reverse hepatic fatty acid accumulation and improve hepatic insulin resistance, providing a new perspective on the role of hypoxia in NAFLD. Moreover, our study highlights the importance of lipid droplet autophagy in the adaptation of hepatocytes to hypoxic conditions. The enhanced lipid droplet autophagy observed in the HFD – 4500 m group may be a compensatory mechanism to cope with the stress of hypoxia and high – fat diet.

Despite these significant findings, our study has several limitations. One major limitation is the experimental conditions. The study was conducted in a mouse model, which may not fully represent the complex physiological and pathological conditions of human NAFLD. There are significant differences in metabolism, immune response, and genetic background between mice and humans, which may limit the generalizability of our results to human populations. Another limitation is that we did not fully consider the differences in dietary structure between high-altitude and low-altitude areas, particularly the variations in carbohydrate, fat composition, and micronutrient intake, which could significantly influence the incidence of NAFLD.

In conclusion, while our study provides valuable insights into the regulatory mechanisms of NAFLD under hypoxic conditions, further research is needed to overcome its limitations. Large-scale epidemiological surveys and simulation experiments should be conducted in high-altitude regions to deeply explore the combined effects of diet and altitude on NAFLD, thereby providing a scientific basis for formulating targeted public health interventions.

## Conclusions

This study revealed that hypoxic conditions enhanced lipid droplet autophagy, reduced glycolipid accumulation, and alleviated mitochondrial damage in NAFLD mice.

## Author contributions

**Conceptualization:** Meiyuan Tian, Zhe Liu, Li Sun, Yuan Jiang.

**Data curation:** Na Zhao, Jing Hou.

**Formal analysis:** Na Zhao.

**Funding acquisition:** Yuan Jiang.

**Investigation:** Dengliang Huang.

**Methodology:** Meiyuan Tian, Yaogang Zhang, Zhe Liu.

**Software:** Guangcun Zhang.

**Supervision:** Guangcun Zhang.

**Writing – original draft:** Meiyuan Tian.

**Writing – review & editing:** Meiyuan Tian, Ma Yanyan.

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
