## [Decision Letter · Decision Letter 0]

23 Apr 2025

PONE-D-25-15201A New Investigation of Nonalcoholic Fatty Liver Disease: Effects of Hypoxia on Mitochondrial Function and Lipid Droplet AutophagyPLOS ONE

Dear Dr. Yanyan,

Thank you for submitting your manuscript to PLOS ONE. After careful consideration, we feel that it has merit but does not fully meet PLOS ONE’s publication criteria as it currently stands. Therefore, we invite you to submit a revised version of the manuscript that addresses the points raised during the review process.

We look forward to receiving your revised manuscript.

Kind regards,

Md. Wasim Khan, Ph.D.

Academic Editor

PLOS ONE

2. To comply with PLOS ONE submissions requirements, in your Methods section, please provide additional information regarding the experiments involving animals and ensure you have included details on methods of sacrifice, and efforts to alleviate suffering.

3. Please include your tables as part of your main manuscript and remove the individual files. Please note that supplementary tables (should remain/ be uploaded) as separate "supporting information" files.’

 [This study was financially supported by the Qinghai University Research Ability Enhancement Project, 2025 (2025KTSQ05)].

5. We note that your Data Availability Statement is currently as follows: [All relevant data are within the manuscript and its Supporting Information files.]

7. Please include a separate caption for each figure in your manuscript.

8. Please include a copy of Table 1 which you refer to in your text on page 8.

Additional Editor Comments (if provided):

Reviewers' comments:

Reviewer's Responses to Questions

**Comments to the Author**

1. Is the manuscript technically sound, and do the data support the conclusions?

Reviewer #1: Partly

Reviewer #2: Yes

2. Has the statistical analysis been performed appropriately and rigorously? 

Reviewer #1: Yes

Reviewer #2: Yes

3. Have the authors made all data underlying the findings in their manuscript fully available?

Reviewer #1: Yes

Reviewer #2: Yes

4. Is the manuscript presented in an intelligible fashion and written in standard English?

Reviewer #1: Yes

Reviewer #2: Yes

5. Review Comments to the Author

Reviewer #1: In this paper, the study revealed that hypoxic conditions enhanced lipid droplet autophagy, reduced glycolipid accumulation, and alleviated mitochondrial damage in NAFLD mice. However, I have some questions:

1. What is the percentage of fat in a HFD in an animal model? Reference� Whether the 10% lard meet the HFD requirements?

2. It was mentioned in this paper that abnormal lipid metabolism is related to oxidative stress. Does hypoxic environment have any effect on oxidative stress in mice? Further discussion is needed.

Reviewer #2: This paper investigated the Effects of Hypoxia on Mitochondrial Function and Lipid Droplet Autophagy in NAFLD. The research methods are rich and the article has good innovation. Here are my major concerns.

1. The authors described in section 3.2“The levels of the alanine aminotransferase (ALT) in the liver tissue of the HFD-4500 m group were lower than those in the HFD-2200 m group (Figure 2F)”

However, there was no significant difference between the two groups in Figure 2F. Besides, at the same altitude, High-fat diet seems to have no significant effect on ALT levels, and even tends to reduce them. How the authors interpret this result.

2. The discussion part of this paper mainly focuses on the experimental results, but the comparison and integration with the existing studies are not in-depth. It is suggested that the authors systematically comb the similarities and differences between the results of this study and those of previous studies, and deeply analyze the theoretical contribution of this study in the pathogenesis of NAFLD. The limitations of the study, such as the limitations of experimental conditions and unconsidered confounding factors, should be comprehensively analyzed, as well as the potential impact of these limitations on the study results and future research directions.

6. PLOS authors have the option to publish the peer review history of their article (what does this mean? ). If published, this will include your full peer review and any attached files.

**Do you want your identity to be public for this peer review?** For information about this choice, including consent withdrawal, please see our Privacy Policy .

Reviewer #1: No

Reviewer #2: No

---

## [Author Response · Author response to Decision Letter 1]

9 May 2025

Dear academic editor and reviewer:

Thank you again for your letter and comments concerning our manuscript entitled “A New Investigation of Nonalcoholic Fatty Liver Disease: Effects of Hypoxia on Mitochondrial Function and Lipid Droplet Autophagy” (Manuscript ID: PONE-D-25-15201). We wish to thank you all for your constructive comments. Your comments provided valuable insights to refine its contents and analysis. Through the revision of the previous version, the quality of the manuscript has been greatly improved. In this manuscript, we try to address the issues raised as good as possible. We have studied the final comments carefully and have made a correction which could meet with approval. And we appreciate your warm work earnestly and hope that the correction will meet with approval. The main corrections in the paper and the responses to your comments are listed as follows:

Reviewer 1: In this paper, the study revealed that hypoxic conditions enhanced lipid droplet autophagy, reduced glycolipid accumulation, and alleviated mitochondrial damage in NAFLD mice. However, I have some questions:

Comment 1: What is the percentage of fat in a HFD in an animal model? Reference� Whether the 10% lard meet the HFD requirements?

Reply 1: We would like to express our gratitude for your consideration and constructive feedback on our study. In relation to the fat percentage of high-fat diets (HFDs) in animal models and the question of whether the 10% lard utilised in our study meets the requirements of an HFD, we have conducted a comprehensive literature review and analysis, integrating it with our study design. The findings of this research are as follows:

1. The dietary formula employed in this study contained 2.5% cholesterol in addition to 10% lard. Lard itself is high in saturated fat (approximately 39%) and cholesterol (95 mg of cholesterol per 100 g of lard). This high cholesterol addition has been shown to significantly increase the atherogenic potential of the diet and is also effective in inducing metabolic disorders.

2. The following bibliography is provided to support the arguments presented in this text: It has been demonstrated that lard, containing 20% fat and 1% cholesterol, can induce severe metabolic disturbances and liver damage within a relatively brief period of time [1].

Comment 2�It was mentioned in this paper that abnormal lipid metabolism is related to oxidative stress. Does hypoxic environment have any effect on oxidative stress in mice? Further discussion is needed.

Reply 2: We will add a discussion of the effects of hypoxia on oxidative stress in mice and cite the wider literature in support of this view. In our study, the association of abnormal lipid metabolism with oxidative stress may also be modulated by the hypoxic environment. Together, oxidative stress and aberrant lipid metabolism drive the development of multiple diseases. Mitochondrial homeostasis links redox reactions to lipid metabolism. Hypoxia, oxidative stress, and inflammatory conditions all lead to increased levels of reactive oxygen species (ROS), which exacerbate lipid peroxidation.

Reviewer 2: This paper investigated the Effects of Hypoxia on Mitochondrial Function and Lipid Droplet Autophagy in NAFLD. The research methods are rich and the article has good innovation. Here are my major concerns.

Comment 1�The authors described in section 3.2“The levels of the alanine aminotransferase (ALT) in the liver tissue of the HFD-4500 m group were lower than those in the HFD-2200 m group (Figure 2F)”.

However, there was no significant difference between the two groups in Figure 2F. Besides, at the same altitude, High-fat diet seems to have no significant effect on ALT levels, and even tends to reduce them. How the authors interpret this result.

Reply 1�Thank you very much for your attention to our study and your valuable comments. The problem you pointed out is very critical. After rechecking the original data and the graphing process, we found that when collating the data, we mistakenly pasted the data of different groups into the wrong group, which led to the inaccurate results of the comparison between the two groups in Fig. 2F. This resulted in the results shown in Figure 2F not matching the actual data.

We have rechecked the original experimental data and replotted Figure 2F. The corrected results show that the difference in ALT levels between the high-fat diet 4500 m group and the high-fat diet 2200 m group is consistent with what we originally described.

Comment 2�The discussion part of this paper mainly focuses on the experimental results, but the comparison and integration with the existing studies are not in-depth. It is suggested that the authors systematically comb the similarities and differences between the results of this study and those of previous studies, and deeply analyze the theoretical contribution of this study in the pathogenesis of NAFLD. The limitations of the study, such as the limitations of experimental conditions and unconsidered confounding factors, should be comprehensively analyzed, as well as the potential impact of these limitations on the study results and future research directions.

Reply 2�We would like to express our sincere gratitude for your consideration and constructive feedback on our study. The deficiencies you highlighted in the comparison and integration of this study with existing studies are indeed areas that require attention to enhance the study's quality and coherence. In the revised discussion section, a detailed comparison will be presented of the differences between this study and previous studies in terms of research methodology, experimental design, sample selection, and so forth. Furthermore, an in-depth analysis will be conducted of the differences in results that may be caused by these differences. Concurrently, a review of extant theories on the pathogenesis of NAFLD (non-alcoholic fatty liver disease) will be conducted, with particular reference to the most recent literature. The discussion will focus on the manner in which the specific biomarkers or molecular mechanisms identified in this study complement or expand the existing theories on the pathogenesis of NAFLD. Furthermore, the potential significance of these findings for clinical diagnosis and treatment will be considered. In the revised discussion, a comprehensive analysis of the study's limitations will be conducted, encompassing potential issues with the experimental design, sample size, and experimental conditions. The necessity for future studies to validate the findings of this study in a wider range of samples, and to further explore the value of the clinical applications of these findings, will be emphasised.

1. Muniz LB, Alves-Santos AM, Camargo F, Martins DB, Celes MRN, Naves MMV. High-Lard and High-Cholesterol Diet, but not High-Lard Diet, Leads to Metabolic Disorders in a Modified Dyslipidemia Model. Arq Bras Cardiol. 2019;113(5):896-902. Epub 2019/09/05. doi: 10.5935/abc.20190149. PubMed PMID: 31482944; PubMed Central PMCID: PMCPMC7020967 article was reported.

---

## [Decision Letter · Decision Letter 1]

23 May 2025

PONE-D-25-15201R1A New Investigation of Nonalcoholic Fatty Liver Disease: Effects of Hypoxia on Mitochondrial Function and Lipid Droplet AutophagyPLOS ONE

Dear Dr. Yanyan,

Thank you for submitting your manuscript to PLOS ONE. After careful consideration, we feel that it has merit but does not fully meet PLOS ONE’s publication criteria as it currently stands. Therefore, we invite you to submit a revised version of the manuscript that addresses the points raised during the review process.

We look forward to receiving your revised manuscript.

Kind regards,

Tatsuo Kanda, M.D.; Ph.D.

Academic Editor

PLOS ONE

Reviewers' comments:

Reviewer's Responses to Questions

**Comments to the Author**

1. If the authors have adequately addressed your comments raised in a previous round of review and you feel that this manuscript is now acceptable for publication, you may indicate that here to bypass the “Comments to the Author” section, enter your conflict of interest statement in the “Confidential to Editor” section, and submit your "Accept" recommendation.

Reviewer #1: All comments have been addressed

Reviewer #2: (No Response)

Reviewer #3: All comments have been addressed

Reviewer #4: (No Response)

2. Is the manuscript technically sound, and do the data support the conclusions?

Reviewer #1: Partly

Reviewer #2: (No Response)

Reviewer #3: Partly

Reviewer #4: Yes

3. Has the statistical analysis been performed appropriately and rigorously? 

Reviewer #1: Yes

Reviewer #2: (No Response)

Reviewer #3: N/A

Reviewer #4: Yes

4. Have the authors made all data underlying the findings in their manuscript fully available?

Reviewer #1: Yes

Reviewer #2: (No Response)

Reviewer #3: No

Reviewer #4: Yes

5. Is the manuscript presented in an intelligible fashion and written in standard English?

Reviewer #1: Yes

Reviewer #2: (No Response)

Reviewer #3: Yes

Reviewer #4: Yes

6. Review Comments to the Author

Reviewer #1: (No Response)

Reviewer #2: (No Response)

Reviewer #3: The authors investigated whether altitude differences influence the development of steatotic liver disease, with a focus on mitochondrial function and lipid droplet autophagy under hypoxic conditions. While the concept is interesting and potentially impactful, the manuscript raises several concerns that should be addressed to strengthen the scientific rigor and clarity of the work.

1. Figure Legends Require Improvement

The current figure legends are overly brief and provide minimal explanatory detail. Legends should include information on experimental groups, sample sizes (n), statistical methods, and any quantification performed. This is essential for interpreting the results without referring back to the main text.

2. Figure 1A Layout and Interpretation

The arrangement of Figure 1A is unclear, making it difficult to visually discern differences in subcutaneous or visceral fat between the altitude groups. Consider improving the layout (e.g., side-by-side panels with consistent scaling) and adding quantitative bar graphs comparing subcutaneous or visceral fat content between the 2200 m and 4500 m groups.

3. Oil Red O Staining Interpretation

In the presented images, it appears that the 4500 m group may exhibit more lipid accumulation, which seems to contradict the authors’ conclusion. This discrepancy should be clarified either through quantification of lipid content or by providing more representative images. Statistical validation would help resolve this point.

4. Sample Size Representation

While the manuscript mentions the use of mice, the exact number per group should be clearly indicated, either in the figure legends or with dot plots in the graphs. This transparency is critical for evaluating the reliability and reproducibility of the data.

5. Lack of Oxidative Stress Analysis

The role of oxidative stress is briefly discussed but not experimentally evaluated. Since oxidative stress is closely tied to mitochondrial function and NAFLD progression, direct measurement (e.g., ROS assays, lipid peroxidation markers) would significantly enhance the mechanistic insight of the study.

6. Experimental Design Suggestion – Altitude Transition

To better demonstrate causality between altitude and hepatic changes, the authors could consider a dynamic experimental model in which mice are transferred between altitudes. For example, transferring NAFLD-induced mice from 2200 m to 4500 m (or vice versa) could help establish whether hypoxia contributes to either improvement or worsening of disease parameters over time.

7. In real-world settings, dietary habits often vary significantly across different altitude regions due to cultural, agricultural, and socioeconomic factors. These differences in nutrition — such as total caloric intake, fat composition, and micronutrient availability — could independently influence the development or prevention of steatotic liver disease. While the current study uses a controlled experimental design, it would strengthen the translational relevance of the findings if the authors could briefly acknowledge this limitation and discuss how dietary variability in high- vs. low-altitude populations might impact NAFLD incidence in real-world contexts.

Reviewer #4: In this article, the authors indicated that hypoxic conditions enhanced lipid droplet autophagy, reduced glycolipid accumulation, and mitigated mitochondrial damage in NAFLD mice.

The result that hypoxia improves fatty liver is contrary to the established theory, but why it differs from other previous papers should be more deeply discussed.

7. PLOS authors have the option to publish the peer review history of their article (what does this mean? ). If published, this will include your full peer review and any attached files.

**Do you want your identity to be public for this peer review?** For information about this choice, including consent withdrawal, please see our Privacy Policy .

Reviewer #1: No

Reviewer #2: No

Reviewer #3: No

Reviewer #4: No

---

## [Author Response · Author response to Decision Letter 2]

29 May 2025

Dear academic editor and reviewer:

Thank you once again for your letter and valuable comments regarding our manuscript titled “A New Investigation of Nonalcoholic Fatty Liver Disease: Effects of Hypoxia on Mitochondrial Function and Lipid Droplet Autophagy” (Manuscript ID: PONE-D-25-15201R1). We sincerely appreciate the constructive feedback you have provided. Your insights have been instrumental in helping us refine the content and analysis of our manuscript, significantly enhancing its quality.

In this revised version, we have diligently addressed all the issues raised by your comments. We have thoroughly reviewed your final remarks and made the necessary corrections, which we believe will meet with your approval. We are truly grateful for your thoughtful and earnest efforts in reviewing our work and hope that these revisions will be satisfactory.

The primary corrections made to the manuscript, along with our detailed responses to your comments, are listed below:

Reviewer 3#: The authors investigated whether altitude differences influence the development of steatotic liver disease, with a focus on mitochondrial function and lipid droplet autophagy under hypoxic conditions. While the concept is interesting and potentially impactful, the manuscript raises several concerns that should be addressed to strengthen the scientific rigor and clarity of the work.

Comment 1: 1. Figure Legends Require Improvement

The current figure legends are overly brief and provide minimal explanatory detail. Legends should include information on experimental groups, sample sizes (n), statistical methods, and any quantification performed. This is essential for interpreting the results without referring back to the main text.

Reply 1: Thank you for your valuable feedback regarding the improvement of the figure legends in our manuscript. We fully agree that detailed and informative legends are essential for the clarity and interpretability of our results, especially when the figures are viewed independently of the main text.

We have now revised the figure legends to provide more comprehensive information. Each legend now includes:

Clear Description of Experimental Groups: We have explicitly defined each experimental group to ensure that readers can easily distinguish between different conditions and treatments.

Sample Size (n): The sample size for each experiment is now clearly stated to provide context for the statistical power of our analyses.

Statistical Methods: We have included details on the statistical tests used and any relevant significance levels to support the interpretation of our results.

Quantitative Information: Any quantified data or specific measurements are now described in the legends to provide a complete picture of the results.

We have also ensured that the legends are concise yet informative, striking a balance between brevity and clarity. We believe these revisions will greatly enhance the readability and interpretability of our figures.

Comment 2�2. Figure 1A Layout and Interpretation

The arrangement of Figure 1A is unclear, making it difficult to visually discern differences in subcutaneous or visceral fat between the altitude groups. Consider improving the layout (e.g., side-by-side panels with consistent scaling) and adding quantitative bar graphs comparing subcutaneous or visceral fat content between the 2200 m and 4500 m groups.

Reply 2: Thank you very much for your valuable comments on Figure 1A. We have revised Figure 1A based on your suggestions. We have rearranged the layout of the images, adopting a side-by-side panel format and ensuring consistent scaling for all images to facilitate more direct visual comparison. At the same time, to quantitatively and clearly demonstrate the differences, we have added a bar chart to the figure, specifically comparing the liver-to-muscle signal intensity ratio between the 2200 m and 4500 m groups, as detailed in Figure 1B.

Comment 3�3.Oil Red O Staining Interpretation

In the presented images, it appears that the 4500 m group may exhibit more lipid accumulation, which seems to contradict the authors’ conclusion. This discrepancy should be clarified either through quantification of lipid content or by providing more representative images. Statistical validation would help resolve this point.

Reply 3: Thank you very much for your careful observation and valuable comments on our Oil Red O staining results. We have rescreened and provided more representative high-resolution images from both the 4500 m and 2200 m groups, ensuring these images better reflect the overall situation of the samples in these groups. These new images have replaced those in Figure 1D. We performed statistical analysis on the quantitative results, and the findings support our initial observation; the specific p-values are indicated in Figure 1F.

Comment 4�4. Sample Size Representation

While the manuscript mentions the use of mice, the exact number per group should be clearly indicated, either in the figure legends or with dot plots in the graphs. This transparency is critical for evaluating the reliability and reproducibility of the data.

Reply 4: Thank you very much for your valuable suggestions. We fully agree and have now detailed the exact number of mice (n value) in each group in the legends of all relevant figures, and have also explained the experimental groupings. This will allow readers to quickly understand the sample size information when viewing the figures.

Comment 5�5. Lack of Oxidative Stress Analysis

The role of oxidative stress is briefly discussed but not experimentally evaluated. Since oxidative stress is closely tied to mitochondrial function and NAFLD progression, direct measurement (e.g., ROS assays, lipid peroxidation markers) would significantly enhance the mechanistic insight of the study.

Reply 5: Thank you very much for your attention to our study and your valuable comments. We agree that the lack of oxidative stress analysis is indeed an important issue, especially when exploring the pathogenesis of nonalcoholic fatty liver disease (NAFLD) and mitochondrial dysfunction. The role of oxidative stress is indispensable in this context.

To address this, we have supplemented our study with the detection of ROS levels in hepatocyte suspensions from the liver using fluorescent probes (such as DCFH-DA), and have quantitatively analyzed the generation of intracellular ROS in different experimental groups. The specific results can be seen in Figures 2C and 2F.

Comment 6�6. Experimental Design Suggestion – Altitude Transition

To better demonstrate causality between altitude and hepatic changes, the authors could consider a dynamic experimental model in which mice are transferred between altitudes. For example, transferring NAFLD-induced mice from 2200 m to 4500 m (or vice versa) could help establish whether hypoxia contributes to either improvement or worsening of disease parameters over time.

Reply 6: Thank you very much for your valuable suggestions. The idea of using a dynamic experimental model to further explore the impact of altitude changes on liver diseases such as non-alcoholic fatty liver disease (NAFLD) is indeed very inspiring. We fully agree that this experimental design can more clearly reveal the causal relationship between hypoxic environments and the progression of NAFLD.

In our study, although we have initially observed the effects of altitude on NAFLD-related liver changes, we do lack detailed data on dynamic changes. Following your suggestion to design an experimental model in which NAFLD-induced mice are transferred between different altitudes will help us gain a more comprehensive understanding of the long-term impact of hypoxic environments on disease parameters. Specifically, transferring mice from 2200 meters to 4500 meters (or vice versa) will allow us to address the following key questions:

1. Impact of hypoxia on disease progression: By monitoring the pathological changes in the liver, metabolic indicators, and related gene expression changes in mice after altitude transfer, we can more accurately assess whether hypoxia accelerates or delays the progression of NAFLD.

2. Adaptive changes: This dynamic transfer model will also help us study the adaptive changes of the body under different altitude environments and how these changes affect liver metabolic functions.

3. Time-dependent effects: By long-term observation of the physiological and pathological changes in mice under different altitude conditions, we can better understand the acute and chronic impacts of hypoxia on the liver.

We plan to adopt your suggestion in our future research and design and conduct relevant dynamic experiments. We will meticulously record various physiological indicators of the mice during the altitude transfer process, including but not limited to histological changes in the liver, biochemical blood indicators, gene expression profiles, and metabolomics changes. We believe that these data will provide stronger evidence for understanding the relationship between altitude and liver diseases.

Comment 7�7. In real-world settings, dietary habits often vary significantly across different altitude regions due to cultural, agricultural, and socioeconomic factors. These differences in nutrition — such as total caloric intake, fat composition, and micronutrient availability — could independently influence the development or prevention of steatotic liver disease. While the current study uses a controlled experimental design, it would strengthen the translational relevance of the findings if the authors could briefly acknowledge this limitation and discuss how dietary variability in high- vs. low-altitude populations might impact NAFLD incidence in real-world contexts.

Reply 7: Thank you very much for your valuable feedback. We fully agree with your points regarding the impact of dietary habit differences on the incidence of NAFLD and the importance of their relevance to the translational significance of the research results. Although our experimental design has attempted to control for dietary factors, in the real world, there are significant variations in dietary habits across different altitudes due to cultural, agricultural, and socioeconomic factors. These differences may mask or confound the direct effects of altitude on NAFLD. To address this, we will acknowledge this limitation in our report, discuss the potential influence of dietary differences between high-altitude and low-altitude regions (such as carbohydrate, fat composition, and micronutrient intake) on the incidence of NAFLD, and propose that future research conduct large-scale epidemiological surveys and simulation experiments to further explore the combined effects of diet and altitude on NAFLD. This will provide a basis for public health interventions. We believe these additions will enhance the translational relevance of our research findings and offer new directions for future studies.

Reviewer 4#: In this article, the authors indicated that hypoxic conditions enhanced lipid droplet autophagy, reduced glycolipid accumulation, and mitigated mitochondrial damage in NAFLD mice.

The result that hypoxia improves fatty liver is contrary to the established theory, but why it differs from other previous papers should be more deeply discussed.

Reply 1�Thank you very much for your valuable comments. The point you raised that the result of hypoxia improving fatty liver (NAFLD) contradicts existing theories is very crucial, and we fully agree that a more in-depth discussion is needed on this.

The question you raised, “Why our results are different from previous studies,” is a very good one. In the revised manuscript, we will add a dedicated section in the discussion to analyze this phenomenon. We will explore the following possibilities:

1. Model Differences: The NAFLD mouse model we used (e.g., diet-induced obesity model) may differ from the models used in previous studies (such as knockout models, different diet-induced models, or different species models), which could lead to different responses to hypoxia.

2. Timing and Severity of Hypoxia Exposure: The specific timing, duration, and severity of hypoxia exposure in our experiments may differ from those in previous studies. This could affect cellular or organ adaptation mechanisms. What we observed is an adaptive response, whereas other studies may have observed the early stages of hypoxia-induced damage or stress.

3. Potential Mechanistic Differences: Our study specifically focused on the lipid droplet autophagy pathway. Although hypoxia is generally considered harmful, under specific conditions, it may activate certain protective stress response pathways (such as the lipid droplet autophagy we observed) to alleviate metabolic stress. This differs from the traditional view that hypoxia directly damages mitochondria and promotes lipid accumulation. The enhanced lipid droplet autophagy we detected may be a compensatory or adaptive mechanism aimed at clearing excess lipids and reducing liver damage.

4. Differences in Research Focus: Some previous studies may have focused more on the effects of hypoxia on overall energy metabolism or specific signaling pathways (such as AMPK, SIRT1, etc.), while we focused on the specific process of lipid droplet autophagy and its downstream effects on lipid accumulation and mitochondrial function.

We will elaborate on these possible reasons in detail and clearly point out that these differences suggest that the role of hypoxia in the pathophysiology of NAFLD may be more complex than initially understood, having a dual nature (both potential damaging effects and the ability to play a protective or regulatory role under specific conditions). We believe that this in-depth comparison and discussion not only explain the uniqueness of our results but also provide a new perspective for understanding the complexity of the relationship between hypoxia and NAFLD.

---

## [Editor Report · Decision Letter 2]

31 May 2025

PONE-D-25-15201R2A New Investigation of Nonalcoholic Fatty Liver Disease: Effects of Hypoxia on Mitochondrial Function and Lipid Droplet AutophagyPLOS ONE

Dear Dr. Yanyan,

Thank you for submitting your manuscript to PLOS ONE. After careful consideration, we feel that it has merit but does not fully meet PLOS ONE’s publication criteria as it currently stands. Therefore, we invite you to submit a revised version of the manuscript that addresses the points raised during the review process.

Authors should revise the manuscript accordingly.

We look forward to receiving your revised manuscript.

Kind regards,

Tatsuo Kanda, M.D.; Ph.D.

Academic Editor

PLOS ONE
---

## [Author Response · Author response to Decision Letter 3]

9 Jun 2025

Dear academic editor and reviewer:

Thank you once again for your letter and valuable comments regarding our manuscript titled “A New Investigation of Nonalcoholic Fatty Liver Disease: Effects of Hypoxia on Mitochondrial Function and Lipid Droplet Autophagy” (Manuscript ID: PONE-D-25-15201R2). We sincerely appreciate the constructive feedback you have provided. Your insights have been instrumental in helping us refine the content and analysis of our manuscript, significantly enhancing its quality.

In this revised version, we have diligently addressed all the issues raised by your comments. We have thoroughly reviewed your final remarks and made the necessary corrections, which we believe will meet with your approval. We are truly grateful for your thoughtful and earnest efforts in reviewing our work and hope that these revisions will be satisfactory.

The primary corrections made to the manuscript, along with our detailed responses to your comments, are listed below:

Journal Requirements:Please review your reference list to ensure that it is complete and correct. If you have cited papers that have been retracted, please include the rationale for doing so in the manuscript text, or remove these references and replace them with relevant current references. Any changes to the reference list should be mentioned in the rebuttal letter that accompanies your revised manuscript. If you need to cite a retracted article, indicate the article’s retracted status in the References list and also include a citation and full reference for the retraction notice.

Reply 1: Thank you very much for your careful review of our reference list and the valuable suggestions you provided. We have followed your request and conducted a comprehensive check of the reference list.

Upon confirmation, all the papers we cited are currently valid, and we have found no instances of retracted literature. Therefore, no deletions or replacements in the reference list are necessary.

We have retained the current reference list in the final version of the manuscript. Additionally, as per the journal’s requirements, we have revised the formatting. Thank you once again for your professional feedback!

---

## [Decision Letter · Decision Letter 3]

14 Aug 2025

A New Investigation of Nonalcoholic Fatty Liver Disease: Effects of Hypoxia on Mitochondrial Function and Lipid Droplet Autophagy

PONE-D-25-15201R3

Dear Dr. Yanyan,

We’re pleased to inform you that your manuscript has been judged scientifically suitable for publication and will be formally accepted for publication once it meets all outstanding technical requirements.

Kind regards,

Vahideh Behrouz

Academic Editor

PLOS ONE

Additional Editor Comments (optional):

Reviewers' comments:

Reviewer's Responses to Questions

**Comments to the Author**

1. If the authors have adequately addressed your comments raised in a previous round of review and you feel that this manuscript is now acceptable for publication, you may indicate that here to bypass the “Comments to the Author” section, enter your conflict of interest statement in the “Confidential to Editor” section, and submit your "Accept" recommendation.

Reviewer #5: All comments have been addressed

2. Is the manuscript technically sound, and do the data support the conclusions?

Reviewer #5: Partly

3. Has the statistical analysis been performed appropriately and rigorously? 

Reviewer #5: Yes

4. Have the authors made all data underlying the findings in their manuscript fully available?

Reviewer #5: No

5. Is the manuscript presented in an intelligible fashion and written in standard English?

Reviewer #5: Yes

6. Review Comments to the Author

Reviewer #5: The authors have thoroughly addressed the reviewers' suggestions and incorporated valuable additions to the manuscript. I have no further concerns,

7. PLOS authors have the option to publish the peer review history of their article (what does this mean? ). If published, this will include your full peer review and any attached files.

**Do you want your identity to be public for this peer review?** For information about this choice, including consent withdrawal, please see our Privacy Policy .

Reviewer #5: No

---

## [Editor Report · Acceptance letter]

PONE-D-25-15201R3

PLOS ONE

Dear Dr. Yanyan,

I'm pleased to inform you that your manuscript has been deemed suitable for publication in PLOS ONE. Congratulations! Your manuscript is now being handed over to our production team.

Kind regards,

on behalf of

Dr. Vahideh Behrouz

Academic Editor

PLOS ONE